# An Empirical Study of Self-Supervised Learning with Wasserstein Distance

**DOI:** 10.3390/e26110939

**Published:** 2024-10-31

**Authors:** Makoto Yamada, Yuki Takezawa, Guillaume Houry, Kira Michaela Düsterwald, Deborah Sulem, Han Zhao, Yao-Hung Tsai

**Affiliations:** 1Machine Learning and Data Science Unit, Okinawa Institute of Science and Technology, Okinawa 904-0412, Japan; yuki-takezawa@ml.ist.i.kyoto-u.ac.jp (Y.T.); guillaume.houry@live.fr (G.H.); kira.dusterwald.21@ucl.ac.uk (K.M.D.);; 2Center for Advanced Intelligence Project RIKEN, Tokyo 103-0027, Japan; 3Department of Intelligence Science and Technology, Kyoto University, Kyoto 606-8501, Japan; 4Paris-Saclay Ecole Normale Superieure, 75005 Paris, France; 5Gatsby Computational Neuroscience Unit, University College London, London WC1E 6BT, UK; 6Barcelona School of Economics, Universitat Pompeu Fabra, 08002 Barcelona, Spain; deborah.sulem@usi.ch; 7Department of Computer Science, University of Illinois at Urbana-Champaign, Champaign, IL 61801, USA; hanzhao@illinois.edu; 8Machine Learning Department, School of Computer Science, Carnegie Mellon University, Pittsburgh, PA 15213, USA

**Keywords:** optimal transport, Wasserstein distance, self-supervised learning

## Abstract

In this study, we consider the problem of self-supervised learning (SSL) utilizing the 1-Wasserstein distance on a tree structure (a.k.a., Tree-Wasserstein distance (TWD)), where TWD is defined as the L1 distance between two tree-embedded vectors. In SSL methods, the cosine similarity is often utilized as an objective function; however, it has not been well studied when utilizing the Wasserstein distance. Training the Wasserstein distance is numerically challenging. Thus, this study empirically investigates a strategy for optimizing the SSL with the Wasserstein distance and finds a stable training procedure. More specifically, we evaluate the combination of two types of TWD (total variation and ClusterTree) and several probability models, including the softmax function, the ArcFace probability model, and simplicial embedding. We propose a simple yet effective Jeffrey divergence-based regularization method to stabilize optimization. Through empirical experiments on STL10, CIFAR10, CIFAR100, and SVHN, we find that a simple combination of the softmax function and TWD can obtain significantly lower results than the standard SimCLR. Moreover, a simple combination of TWD and SimSiam fails to train the model. We find that the model performance depends on the combination of TWD and probability model, and that the Jeffrey divergence regularization helps in model training. Finally, we show that the appropriate combination of the TWD and probability model outperforms cosine similarity-based representation learning.

## 1. Introduction

Unsupervised learning is a widely studied topic, and includes autoencoders [1] and variational autoencoders (VAEs) [2]. Self-supervised learning (SSL) algorithms, including SimCLR [3], Bootstrap Your Own Latent (BYOL) [4], MoCo [3,5], SwAV [6], SimSiam [7], and DINO [8], can also be regarded as unsupervised learning methods.

One of the main self-supervised algorithms adopts contrastive learning, in which two data points are systematically generated from a common data source, and lower-dimensional representations are found by maximizing the similarity between the positive pairs while minimizing the similarity between negative pairs. Depending on the context, positive and negative pairs can be defined differently. For example, in SimCLR [3], positive pairs correspond to images generated by independently applying different visual transformations, such as rotation and cropping. In multimodal learning, however, positive pairs are defined as the same examples corresponding in different modalities, such as images and text [9]. The flexibility of formulating positive and negative pairs also makes contrastive learning widely applicable beyond the image domain. This is a powerful pre-training method, because SSL does not require label information and can be trained using several data points.

In addition to contrastive learning-based SSL, non-contrastive approaches, such as BYOL [4], SwAV [6], and SimSiam [7], have been widely used. The fundamental concept of non-contrastive approaches involves the utilization of momentum and/or stop-gradient techniques to prevent mode collapse, as opposed to relying on negative sampling. Many of these approaches employ negative cosine similarity as a loss function. However, a limited number of SSL methods utilize distribution measures, such as cross-entropy, as exemplified by DINO [8], and simplicial embedding [10].

In this paper, leveraging the idea of distribution measures, for the first time we empirically investigate SSL performance using the Wasserstein distance. The Wasserstein distance, a widely adopted optimal transport-based distance for measuring distributional discrepancies, is useful in various machine learning tasks, including generative adversarial networks [11], document classification [12,13], image matching [14], and algorithmic fairness [15,16]. The 1-Wasserstein distance is also known as the earth mover’s distance (EMD) and the word mover’s distance (WMD) [12].

In this study, we consider an SSL framework with a 1-Wasserstein distance under a tree metric (i.e., Tree-Wasserstein distance (TWD)) [17,18]. TWD includes the sliced Wasserstein distance [19,20] and total variation as special cases, and can be represented by the $\ell_{1}$ distance between two vectors. Due to the fact that TWD is given as a non-differentiable function, learning simplicial representations through back-propagation of TWD is challenging. Moreover, because the Wasserstein distance is computed from probability vectors, and several representations of probability vectors exist, it is difficult to determine which is most suitable for SSL training. Hence, we investigate a combination of probability models and the structure of TWD. Specifically, we consider the total variation and ClusterTree for TWD structure and show that the total variation is equivalent to a robust variant of TWD. In terms of the probability representations, we propose the combined use of softmax, an ArcFace-based probability model [21], and simplicial embedding (SEM) [10]. Finally, to stabilize the training, we propose a Jeffrey divergence-based regularization. Through SSL experiments, we find that the standard softmax formulation with back-propagation yields poor results. In particular, the non-contrastive SSL case fails to train the model with a simple combination of the Wasserstein distance and softmax function. For total variation, the ArcFace-based model performs well. By contrast, SEM is suitable for ClusterTree, whereas ArcFace-based models achieve modest performance. Moreover, the proposed regularization significantly outperforms its non-regularized counterparts.

**Contribution:** The contributions of this study are summarized below:We propose to use the tree Wasserstein distance for self-supervised learning including SimCLR and SimSiam for the first time.We investigate the combination of probability models and TWD (total variation and ClusterTree). We find that the ArcFace model with prior information is suited for total variation, while SEM [10] is suited for ClusterTree models.We propose a robust variant of TWD (RTWD) and show that RTWD is equivalent to total variation.We propose the Jeffrey divergence regularization for TWD minimization, and find that the regularization significantly stabilizes training.We demonstrate that the combination of TWD and probability models can obtain better performance in self-supervised training for CIFAR10, STL10, and SVHN compared to the cosine similarity in SimCLR experiments, while the performance of CIFAR100 can be improved further in the future.

## 2. Related Work

The proposed method involves unsupervised representation learning and optimal transport.

**Unsupervised Representation Learning:** Representation learning is an important research topic in machine learning and involves several methods. The autoencoder [1] and its variational version [2] are widely employed in unsupervised representation learning methods. Current mainstream SSL approaches are based on a cross-view prediction framework [22] and contrastive learning has emerged as a prominent SSL paradigm.

In contrastive learning, a model learns by contrasting positive samples (similar instances) with negative samples (dissimilar instances) using methods such as SimCLR [23]. SimCLR employs data augmentation and similarity metrics to encourage the model to project similar instances close together while pushing dissimilar instances apart. This approach has demonstrated efficacy across various domains, including computer vision and natural language processing, thus enabling learning without explicit labels. SimCLR employs the InfoNCE loss [24]. Subsequently to SimCLR, several alternative approaches have been proposed, including the use of Barlow Twins [25]. The Barlow twin loss function attempts to maximize the correlation between positive pairs while minimizing the cross-correlation with negative samples. Barlow Twins is closely related to the Hilbert–Schmidt independence criterion, a kernel-based independence measure [26,27].

One drawback of SimCLR is its reliance on numerous negative samples. To address this issue, recent research has focused on approaches that eliminate the need for negative sampling, such as BYOL [4], SwAV [6], and DINO [8]. For example, BYOL demonstrates training of representations by minimizing the loss between online and target networks. The target network is formed by maintaining a moving average of the online network parameters, and eliminates the need for negative samples. Surprisingly, BYOL showed favorable results compared with SimCLR. SimSiam, introduced by Chen and He [7], utilizes a Siamese network to train the estimation by fixing one of the networks using a stop gradient.

Both of these approaches concentrate on learning low-dimensional representations with real-valued vector embeddings by employing cosine similarity as a similarity measure in contrastive learning. Recently, Lavoie et al. [10] proposed simplicial embedding (SEM), which involves multiple concatenated softmax functions and learns high-dimensional sparse non-negative representations. This innovation significantly enhances classification accuracy.

All of these approaches employ either a negative cosine similarity or cross-entropy as a loss function. In contrast, use of the Wasserstein distance in this context has not been studied.

**Divergence and optimal transport:** Measuring the divergence between two probability distributions is a fundamental research problem in machine learning. It has utility for various downstream applications, including document classification [12,13], image matching [14], and algorithmic fairness [15,16]. One widely adopted divergence measure is Kullback–Leibler (KL) divergence [28]. However, KL divergence can diverge to infinity when the supports of the two input probability distributions do not overlap.

The Wasserstein distance, or, as it is known in the computer vision community, EMD, can handle differences in supports between probability distributions. Another key advantages of the Wasserstein distance over KL is that it can identify matches between the data samples. For example, Sarlin et al. [14] proposed SuperGlue, leveraging optimal transport for correspondence determination in local feature sets.

In NLP, Kusner et al. [12] introduced WMD, a Wasserstein distance pioneer in textual similarity tasks that is widely utilized, including for text generation evaluation [29]. Sato et al. [13] further studied the properties of WMD. Another interesting approach is the word rotator distance (WRD) [30], which utilizes the norm of word vectors as a simplicial representation and significantly improves WMD’s performance. However, these methods incur cubic-order computational costs, rendering them unsuitable for extensive distribution-comparison tasks.

The Wasserstein distance can be computed efficiently via linear programming (cubic complexity). However, to speed up EMD and Wasserstein distance computation, Cuturi [31] introduced the Sinkhorn algorithm, which solves the entropic regularized optimization problem and achieves quadratic order Wasserstein distance computation ($O({\bar{n}}^{2})$), where $\bar{n}$ is the number of data points. Moreover, because the optimal solution from the Sinkhorn algorithm can be obtained using an iterative algorithm, it can be easily incorporated into deep learning applications, making optimal transport widely applicable. One limitation of the Sinkhorn algorithm is that it still requires quadratic-time computation, and the final solution depends highly on the regularization parameter.

An alternative approach is the sliced Wasserstein distance (SWD) [19,20], which solves the optimal transport problem within a projected one-dimensional subspace. The algorithm for Wasserstein distance computation over reals essentially applies sorting as a subroutine; thus, SWD offers $O(\bar{n}\log\bar{n})$ computation. SWD’s extensions include the generalized sliced Wasserstein distance for multidimensional cases [32]; the max-sliced Wasserstein distance, which determines the optimal transport-enhancing 1D subspace [33,34]; and the subspace-robust Wasserstein distance [35].

The 1-Wasserstein distance with a tree metric (also known as the Tree-Wasserstein Distance (TWD)) is a generalization of the sliced Wasserstein distance and total variation [17,18,36]. The TWD is also known as the UniFrac distance [37] and is assumed to have a phylogenetic tree beforehand. An important property of TWD is that TWD has an analytical solution for the L1 distance of tree-embedded vectors.

Originally, TWD was studied in theoretical computer science, known as the QuadTree algorithm [17]. This has recently been extended by the ML community to include unbalanced TWD [38,39], supervised Wasserstein training [40], tree barycenters [41], robust Wasserstein distance [42], unsupervised tree construction [43], and greedy matching [44]. Moreover, graph-based optimal transport has also been studied recently [45,46] and has been used for many applications including natural language processing [47,48] and single-cell analysis [49].

These approaches focus on approximating the 1-Wasserstein distance through tree construction and often utilize constant-edge weights. In terms of approaches that consider non-uniform edge weights, Backurs et al. [50] introduced FlowTree, amalgamating QuadTree and cost matrix methods, outperforming QuadTree. They guaranteed that QuadTree and FlowTree approximate nearest neighbors by employing the 1-Wasserstein distance. Dey and Zhang [51] proposed an L1-embedding for approximating the 1-Wasserstein distance for persistence diagrams. Finally, Yamada et al. [52] proposed a computationally efficient tree weight estimation technique for TWD and empirically demonstrated that TWD can attain comparable performance to the Wasserstein distance, while achieving computational speeds several orders of magnitude faster than linear programming computation of the Wasserstein distance.

Most existing studies on TWD have focused on tree construction [17,18,40] and edge weight estimation [52]. Frogner et al. [53] and Toyokuni et al. [54] considered utilizing the Wasserstein distance for multi-label classification. These studies focused on supervised learning by employing softmax as the probability model. In this study, we investigate the Wasserstein distance by employing an SSL framework and evaluate various probability models.

## 3. Background

### 3.1. Self-Supervised Learning Methods

**SimCLR [3]:** Given *n* input vectors ${\left\{x_{i}\right\}}_{i=1}^{n}$, where $x_{i}\in R^{d}$, define the data transformation functions $u^{\left(1\right)}=\phi_{1}\left(x\right)\in R^{d}$ and $u^{\left(2\right)}=\phi_{2}\left(x\right)\in R^{d}$. In the context of image applications, $u^{\left(1\right)}$ and $u^{\left(2\right)}$ can be understood as two image transformations over a given image: translation, rotation, blurring, etc. The neural network model is denoted as $f_{\theta }\left(u\right)\in R^{d_{\mathrm{out}}}$, where θ is a learnable parameter.

SimCLR attempts to train the model by learning features such that $z^{\left(1\right)}=f_{\theta }\left(u^{\left(1\right)}\right)$ and $z^{\left(2\right)}=f_{\theta }\left(u^{\left(2\right)}\right)$ are close after the feature mapping, while ensuring that both are distant from the feature map of $u^{\prime }$, where $u^{\prime }$ is a negative sample generated from a different input image. To this end, InfoNCE loss [24] is employed in the SimCLR model:$$\begin{array}{cc} \ell_{\mathrm{InfoNCE}}\left(z_{i}^{\left(1\right)},z_{i}^{\left(2\right)}\right) & =-\log\frac{\exp\left(\mathrm{sim}\left(z_{i}^{\left(1\right)},z_{i}^{\left(2\right)}\right)/\tau \right)}{\bar{Z}}\\ \; & =-\mathrm{sim}(z_{i}^{\left(1\right)},z_{i}^{\left(2\right)})/\tau +\log\left(\bar{Z}\right), \end{array}$$
where $\bar{Z}=\sum_{k=1}^{2R}\delta_{k\neq i}\exp\left(\mathrm{sim}\left(z_{i}^{\left(1\right)},{\tilde{z}}_{k}\right)/\tau \right)$ is the normalizer, *R* is the batch size and $\mathrm{sim}(z,z^{\prime })$ is a similarity function that takes a higher positive value when z and $z^{\prime }$ are similar and a smaller (positive or negative) value when z and $z^{\prime }$ are dissimilar. τ is the temperature parameter, and $\delta_{k\neq i}$ is a delta function that takes a value of 1 when k≠i and 0 otherwise. In contrastive learning, we aim to minimize the InfoNCE loss function. To achieve an optimal solution, we need to maximize the similarity $\mathrm{sim}\left(z_{i}^{\left(1\right)},z_{i}^{\left(2\right)}\right)$ while minimizing log(Z). The first term aims to make $z_{i}^{\left(1\right)}$ and $z_{i}^{\left(2\right)}$ as similar as possible. The second term is a log-sum-exp function, which can be interpreted for small τ as
$$\begin{array}{cc} \log(Z) & =\log\left(\sum_{k=1}^{2R}\delta_{k\neq i}\exp\left(\mathrm{sim}\left(z_{i}^{\left(1\right)},{\tilde{z}}_{k}\right)/\tau \right)\right),\\ \; & \approx \max_{k}\left(\mathrm{sim}\left(z_{i}^{\left(1\right)},{\tilde{z}}_{k}\right)\right). \end{array}$$

By minimizing log(Z), we can make $z_{i}^{\left(1\right)}$ dissimilar to the negative samples ${\tilde{z}}_{k}$. Due to the fact that we attempt to minimize the maximum similarity between input $z_{i}$ and its negative samples, we can make $z_{i}$ and its negative samples dissimilar via the second term.

In SimCLR, the parameters are learned by minimizing the InfoNCE loss.
$$\begin{array}{c} \hat{\theta }:=\underset{\theta }{\mathrm{argmin}\;}\sum_{i=1}^{n}\ell_{\mathrm{InfoNCE}}\left(f_{\theta }\left(u_{i}^{\left(1\right)}\right),f_{\theta }\left(u_{i}^{\left(2\right)}\right)\right). \end{array}$$

**SimSiam [7]:** SimSiam is a non-contrastive learning method; it does not use negative sampling to prevent mode collapse. In place of negative sampling, SimSiam employs a stop-gradient method. Specifically, the loss function is given by
$$\begin{array}{cc} \; & L_{SimSiam}\left(\theta \right)=\frac{1}{2}L_{1}\left(\theta \right)+\frac{1}{2}L_{2}\left(\theta \right),\\ \; & L_{1}\left(\theta \right)\;=\;-\frac{1}{n}\sum_{i=1}^{n}\;\frac{h{\left(z_{i}\right)}^{⊤}{\bar{z}}_{i}^{\prime }}{∥h\left(z_{i}\right){∥}_{2}{∥{\bar{z}}_{i}^{\prime }∥}_{2}},\;L_{2}\left(\theta \right)\;=\;-\frac{1}{n}\sum_{i=1}^{n}\;\frac{{\bar{z}}_{i}^{⊤}h\left(z_{i}^{\prime }\right)}{∥{\bar{z}}_{i}{∥}_{2}{∥h\left(z_{i}^{\prime }\right)∥}_{2}}, \end{array}$$
where h(·) is the MLP head, $z_{i}$ is a latent variable, and ${\bar{z}}_{i}=\mathrm{StopGradient}\left(z_{i}\right)$ is a latent variable with a stop gradient.

### 3.2. p-Wasserstein Distance

The *p*-Wasserstein distance between two discrete measures, $\mu =\sum_{i=1}^{\bar{n}}a_{i}\delta_{x_{i}}$ and $\mu^{\prime }=\sum_{j=1}^{\bar{m}}a_{j}^{\prime }\delta_{y_{j}}$ is given by
$$\begin{array}{c} W_{p}\left(\mu ,\mu^{\prime }\right)={\left(\min_{\Pi \in U(\mu ,\mu^{\prime })}\sum_{i=1}^{\bar{n}}\sum_{j=1}^{\bar{m}}\pi_{ij}d{\left(x_{i},y_{j}\right)}^{p}\right)}^{1/p}, \end{array}$$
where $U(\mu ,\mu^{\prime })$ denotes the set of transport plans and $U\left(\mu ,\mu^{\prime }\right)=\left\{\Pi \in R_{+}^{\bar{n}\times \bar{m}}:\Pi 1_{\bar{m}}=a,\Pi^{⊤}1_{\bar{n}}=a^{\prime }\right\}$. The Wasserstein distance can be computed using a linear program. However, because this includes an optimization problem, the computation of Wasserstein distance for each iteration is computationally expensive.

### 3.3. 1-Wasserstein Distance with Tree Metric (Tree-Wasserstein Distance)

Another 1-Wasserstein distance is based on trees [17,18]. The 1-Wasserstein distance between two probability distributions $\mu =\sum_{i=1}^{N_{\mathrm{leaf}}}a_{i}\delta_{x_{i}}$ and $\mu^{\prime }=\sum_{j=1}^{N_{\mathrm{leaf}}}a_{j}^{\prime }\delta_{y_{j}}$ with the tree metric is defined as
(1)$$\begin{array}{c} W_{T}\left(\mu ,\mu^{\prime }\right)=\min_{\Pi \in U(\mu ,\mu^{\prime })}\sum_{i=1}^{N_{\mathrm{leaf}}}\sum_{j=1}^{N_{\mathrm{leaf}}}\pi_{ij}d_{T}\left(x_{i},y_{j}\right), \end{array}$$
where $d_{T}\left(x,y\right)$ is the length of the shortest path between x and y on the tree and $N_{\mathrm{leaf}}$ is the number of leaf nodes. TWD can be further represented by the closed form as follows [18]:(2)$$\begin{array}{c} W_{T}\left(\mu ,\mu^{\prime }\right)=\sum_{e\in E}w_{e}\left|\mu \left(\Gamma \left(v_{e}\right)\right)-\mu^{\prime }\left(\Gamma \left(v_{e}\right)\right)\right|, \end{array}$$
where *e* is an edge index, $w_{e}\in R_{+}$ is the edge weight of edge *e*, $v_{e}$ is the *e*th node index, and $\mu (\Gamma \left(v_{e}\right))$ is the total mass of the subtree with root $v_{e}$. This closed form solution can be further represented as the L1 distance [40]:$$\begin{array}{cc} W_{T}\left(\mu ,\mu^{\prime }\right) & =∥\mathrm{diag}\left(w\right)Ba-\mathrm{diag}\left(w\right)Ba^{\prime }{∥}_{1}, \end{array}$$
where $B\in {\left\{0,1\right\}}^{N_{\mathrm{node}}\times N_{\mathrm{leaf}}}$ is a tree parameter, ${\left[B\right]}_{i,j}=1$ if node *i* is the ancestor node of leaf node *j* and zero otherwise, $N_{\mathrm{node}}$ is the total number of nodes of a tree, and $w\in R_{+}^{N_{\mathrm{node}}}$ is the edge weight.

For illustration, we provide two examples to demonstrate the *B* matrix by considering a tree with a depth of one and a ClusterTree, as shown in Figure 1. If all edge weights $w_{1}=w_{2}=\ldots =w_{N}=\frac{1}{2}$ in the left panel of Figure 1, then the B matrix is given as B=I. By substituting this result into the TWD, we obtain
$$\begin{array}{c} W_{T}\left(\mu ,\mu^{\prime }\right)=\frac{1}{2}∥a-a^{\prime }{∥}_{1}={∥a-a^{\prime }∥}_{\mathrm{TV}}. \end{array}$$

Thus, the total variation is a special case of TWD. In this setting, the shortest-path distance in the tree corresponds to the Hamming distance. Note that Raginsky et al. [55] also assert that the 1-Wasserstein distance with the Hamming metric $d\left(x,y\right)=\delta_{x\neq y}$ is equivalent to the total variation (Proposition 3.4.1 in Raginsky et al. [55]).

The key advantage of the tree-based approach is that the Wasserstein distance is written in closed form, which is computationally efficient. A chain is included as a special case in the tree. Thus, the widely employed sliced Wasserstein distance is also included as a special case of TWD (Figure 2). Moreover, it has been empirically reported that TWD- and Sinkhorn-based approaches perform similarly in multilabel classification tasks [54].

## 4. SSL with 1-Wasserstein Distance

In this section, we first formulate SSL using TWD. We then introduce ArcFace-based probability models and Jeffrey divergence regularization.

### 4.1. SimCLR with Tree Wasserstein Distance

Let a and $a^{\prime }$ be the embedding vectors of x and $x^{\prime }$ (i.e., $1^{⊤}a=1$ and $1^{⊤}a^{\prime }$) with $\mu =\sum_{j}a_{j}\delta_{e_{j}}$ and $\mu^{\prime }=\sum_{j}a_{j}^{\prime }\delta_{e_{j}}$, respectively. Here, $e_{j}$ is the virtual embedding corresponding to $a_{j}$ or $a_{j}^{\prime }$. e is assumed unavailable in the problem setup. The main idea of this paper is to adopt the negative Wasserstein distance between μ and $\mu^{\prime }$ as the similarity score for SimCLR.
$$\begin{array}{cc} \mathrm{sim}(\mu ,\mu^{\prime }) & =-W_{T}\left(\mu ,\mu^{\prime }\right). \end{array}$$

We assume that B and w are given; that is, both the tree structure and weights are known. In particular, we consider the trees shown in Figure 1.

Following the original design of the InfoNCE loss and by substituting the similarity score given by the negative Wasserstein distance, we obtain the following simplified loss function:$$\begin{array}{cc} \hat{\theta } & :=\underset{\theta }{\mathrm{argmin}\;}\sum_{i=1}^{n}\left(W_{T}\left(\mu_{i}^{\left(1\right)},\mu_{i}^{\left(2\right)}\right)/\tau +\log\sum_{k=1}^{2N}\delta_{k\neq i}\exp\left(-W_{T}\left(\mu_{i}^{\left(1\right)},\mu_{k}^{\left(2\right)}\right)/\tau \right)\right), \end{array}$$
where τ>0 is the temperature parameter for the InfoNCE loss. Although we mainly focus on the InfoNCE loss, the proposed negative Wasserstein distance as a measure of similarity can be used in other contrastive losses as well, e.g., the Barlow Twins.

### 4.2. SimSiam with Tree Wasserstein Distance

Here, we consider a combination of SimSiam and TWD. The loss function of the proposed approach is expressed as
$$\begin{array}{cc} \; & L_{\mathrm{TWDSimsiam}}\left(\theta \right)=\frac{1}{2}L_{1}\left(\theta \right)+\frac{1}{2}L_{2}\left(\theta \right),\\ \; & L_{1}\left(\theta \right)\;=\;\frac{1}{n}\sum_{i=1}^{n}\;W_{T}\left(\mu_{i}^{\left(1\right)},{\bar{\mu }}_{i}^{\left(2\right)}\right),\;\;L_{2}\left(\theta \right)\;=\;\frac{1}{n}\sum_{i=1}^{n}\;W_{T}\left({\bar{\mu }}_{i}^{\left(1\right)},\mu_{i}^{\left(2\right)}\right). \end{array}$$

The distinction to the original SimSiam is that our formulation employs the Wasserstein distance, whereas the original formulation uses cosine similarity.

### 4.3. Robust Variant of Tree Wasserstein Distance

In our setup, it is difficult to estimate the tree structure B and edge weight w because the embedding vectors $e_{1},e_{2},\ldots ,e_{d_{\mathrm{out}}}$ are unavailable. To address this problem, we consider a robust estimation of the Wasserstein distance, such as the subspace-robust Wasserstein distance (SRWD) [35], for TWD. The key idea of SRWD is to solve an optimal transport problem in a subspace in which the distance is maximized. In the TWD case, we can consider solving the optimal transport problem for the maximum shortest-path distance. Specifically, for a given B, we propose the robust TWD (RTWD) as follows:$$\begin{array}{c} \mathrm{RTWD}\left(\mu ,\mu^{\prime }\right)=\frac{1}{2}\min_{\Pi \in U(\mu ,\mu^{\prime })}\;\;\max_{w\in B}\;\sum_{i=1}^{N_{\mathrm{leafs}}}\sum_{j=1}^{N_{\mathrm{leafs}}}\pi_{ij}d_{T}\left(e_{i},e_{j}\right), \end{array}$$
where $B=\{w\in R_{+}^{N_{\mathrm{leaf}}}:B^{⊤}w=1\;\mathrm{and}\;w\geq 0\}$, $d_{T}\left(e_{i},e_{j}\right)$ is the shortest-path distance between $e_{i}$ and $e_{j}$, and $e_{i}$ and $e_{j}$ are embedded in a tree T. This constraint implies that the weights of the ancestor node of leaf node *j* are non-negative and sum to one.

**Proposition** **1.** 
*The robust variant of TWD (RTWD) is equivalent to total variation:*

$$\begin{array}{c} \mathrm{RTWD}\left(\mu ,\mu^{\prime }\right)={∥a-a^{\prime }∥}_{\mathrm{TV}}, \end{array}$$

*where $∥a-a^{\prime }{∥}_{\mathrm{TV}}=\frac{1}{2}{∥a-a^{\prime }∥}_{1}$ denotes the total variation.*


**Proof.** Let $B\in {\left\{0,1\right\}}^{N\times N_{\mathrm{leaf}}}=\left[b_{1},b_{2},\ldots ,b_{N_{\mathrm{leaf}}}\right]$ and $b_{i}\in {\left\{0,1\right\}}^{N}$. The shortest-path distance between leaves *i* and *j* can be represented as [52]
$$\begin{array}{cc} d_{T}\left(e_{i},e_{j}\right) & =w^{⊤}\left(b_{i}+b_{j}-2b_{i}\circ b_{j}\right). \end{array}$$That is, $d_{T}\left(e_{i},e_{j}\right)$ is represented by a linear function with respect to w for a given B and the constraints on w and Π are convex. Thus, strong duality holds, and we obtain the following representation using the minimax theorem [56,57]:
$$\begin{array}{cc} \mathrm{RTWD}(\mu ,\mu^{\prime })\; & =\frac{1}{2}\max_{w\;s.t.\;B^{⊤}w=1\;\mathrm{and}\;w\geq 0}\;\;\min_{\Pi \in U(a,a^{\prime })}\sum_{i=1}^{N_{\mathrm{leafs}}}\sum_{j=1}^{N_{\mathrm{leafs}}}\pi_{ij}w^{⊤}\left(b_{i}+b_{j}-2b_{i}\circ b_{j}\right)\\ \; & =\frac{1}{2}\max_{w\;s.t.\;B^{⊤}w=1\;\mathrm{and}\;w\geq 0}{∥\mathrm{diag}\left(w\right)B\left(a-a^{\prime }\right)∥}_{1}, \end{array}$$
where $\mathrm{TWD}\left(\mu ,\mu^{\prime }\right)=\min_{\Pi \in U(a,a^{\prime })}\sum_{i=1}^{N_{\mathrm{leafs}}}\sum_{j=1}^{N_{\mathrm{leafs}}}\pi_{ij}d_{T}\left(e_{i},e_{j}\right)={∥\mathrm{diag}\left(w\right)B\left(a-a^{\prime }\right)∥}_{1}$.Without loss of generality, we consider $w_{0}=0$. First, we rewrite the norm $∥\mathrm{diag}\left(w\right)B\left(a-a^{\prime }\right){∥}_{1}$ as
$$\begin{array}{c} ∥\mathrm{diag}\left(w\right)B\left(a-a^{\prime }\right){∥}_{1}=\sum_{j=1}^{N}w_{j}|\sum_{k\in \left[N_{\mathrm{leafs}}\right],k\in de\left(j\right)}\left(a_{k}-a_{k}^{\prime }\right)|, \end{array}$$
where de(j) denotes the set of descendants of node j∈[N] (including itself). Using the triangle inequality, we obtain
$$\begin{array}{cc} ∥\mathrm{diag}\left(w\right)B\left(a-a^{\prime }\right){∥}_{1} & \leq \sum_{j=1}^{N}w_{j}\sum_{k\in \left[N_{\mathrm{leafs}}\right],k\in de\left(j\right)}\left|a_{k}-a_{k}^{\prime }\right|\\ \; & =\sum_{k\in [N_{\mathrm{leafs}}]}\left|a_{k}-a_{k}^{\prime }\right|\sum_{j\in [N],j\in pa(k)}w_{j}, \end{array}$$
where pa(k) is the set of ancestors of leaf *k* (including itself). By rewriting the constraint $B^{⊤}w=1$ as $\sum_{j\in [N],j\in pa(k)}w_{j}=1$ for any $k\in [N_{\mathrm{leafs}}]$, we obtain
$$\begin{array}{c} ∥\mathrm{diag}\left(w\right)B\left(a-a^{\prime }\right){∥}_{1}\leq \sum_{k\in [N_{\mathrm{leafs}}]}|a_{k}-a_{k}^{\prime }|=∥a-a^{\prime }{∥}_{1}. \end{array}$$The latter inequality holds for any weight vector w. Therefore, considering the vector such that $w_{j}=1$ if $j\in [N_{\mathrm{leafs}}]$ and 0 otherwise, which satisfies the constraint $B^{⊤}w=1$, we obtain
$$\begin{array}{c} ∥\mathrm{diag}\left(w\right)B\left(a-a^{\prime }\right){∥}_{1}=\sum_{k=1}^{N_{\mathrm{leafs}}}|a_{k}-a_{k}^{\prime }|=∥a-a^{\prime }{∥}_{1}. \end{array}$$This completes the proof of the proposition. □

Based on this proposition, RTWD is equivalent to the total variation and does not depend on the tree structure B. That is, if we do not have prior information about the tree structure, using the total variation is a reasonable choice.

### 4.4. Probability Models

In this section, we discuss several choices of probability models for InfoNCE loss and SimSiam loss.

**Softmax:** The embedded vector with softmax function is given by
$$\begin{array}{c} a_{\theta }\left(x\right)=\mathrm{Softmax}\left(f_{\theta }\left(x\right)\right), \end{array}$$
where $f_{\theta }\left(x\right)$ is a neural network model.

**Simplicial Embedding:** Lavoie et al. [10] proposed a simple yet efficient simplicial embedding method. Assume that the output dimensionality of a neural network model is $d_{\mathrm{out}}$. Then, SEM applies the softmax function to each *V*-dimensional vector of $f_{\theta }\left(x\right)$, where we have $L=d_{\mathrm{out}}/V$ probability vectors. The *ℓ*th softmax function is thus defined as follows:$$\begin{array}{cc} \; & a_{\theta }\left(x\right)={\left[a_{\theta }^{\left(1\right)}{\left(x\right)}^{⊤},a_{\theta }^{\left(2\right)}{\left(x\right)}^{⊤},\ldots ,a_{\theta }^{\left(L\right)}{\left(x\right)}^{⊤}\right]}^{⊤}\\ \; & \mathrm{with}\;\;a_{\theta }^{\left(\ell \right)}\left(x\right)=\mathrm{Softmax}\left(f_{\theta }^{\left(\ell \right)}\left(x\right)\right)/L, \end{array}$$
where $f_{\theta }^{\left(\ell \right)}\left(x\right))\in R^{V}$ is the *ℓ*-th block of a neural network model. We normalize the softmax function by *L* because $a_{\theta }\left(x\right)$ must satisfy the sum-to-one constraint. Note that the softmax function can be regarded as a special case of simplicial embedding (where L=1). In simplicial embedding, the softmax function is applied separately to each subset of the elements. For example, if $d_{\mathrm{out}}=10$ and V=5, the softmax function is applied to each of the two five-dimensional vectors, and the results are then concatenated.

**ArcFace model (AF):** In comparison to SEM, we propose to employ the ArcFace probability model [21]. The ArcFace models employs cosine similarity in addition to softmax.
$$\begin{array}{c} a_{\theta }\left(x\right)=Softmax\left(K^{⊤}f_{\theta }\left(x\right)/\eta \right), \end{array}$$
where $K=\left[k_{1},k_{2},\ldots ,k_{d_{\mathrm{out}}}\right]\in R^{d_{\mathrm{out}}\times d_{\mathrm{prob}}}$ is a learning parameter, $f_{\theta }\left(x\right)$ is the normalized output of a model ($f_{\theta }{\left(x\right)}^{⊤}f_{\theta }\left(x\right)=1$), and η is the temperature parameter. Note that AF has a structure similar to that of transformers [58,59]. The key difference from the original notion of attention in transformers is the normalization of the key matrix K and query vector $f_{\theta }\left(x\right)$.

**AF with Positional Encoding:** To the AF model, one can add one more linear layer and then apply the softmax function; then, the output is similar to the standard softmax function. Here, we propose replacing the key matrix with a normalized positional encoding matrix ($k_{i}^{⊤}k_{i}=1,\forall i$):$$\begin{array}{cc} k_{i} & ={\bar{k}}_{i}/∥{\bar{k}}_{i}{∥}_{2}, \end{array}$$
where ${\bar{k}}_{i}^{\left(2j\right)}=\sin\left(i/{10,000}^{2j/d_{\mathrm{out}}}\right)$ and ${\bar{k}}_{i}^{\left(2j+1\right)}=\cos\left(i/{10,000}^{2j/d_{\mathrm{out}}}\right)$.

**AF with Discrete Cosine Transform Matrix:** Another natural approach would be to utilize an orthogonal matrix as K. Therefore, we propose adopting a discrete cosine transform (DCT) [60] matrix as K, where DCT is in general used for data compression for images. The DCT matrix is expressed as follows [60]:$$\begin{array}{c} k_{i}^{\left(j\right)}=\left\{\begin{array}{cc} 1/\sqrt{d_{\mathrm{out}}} & \left(i=0\right)\\ \sqrt{\frac{2}{d_{\mathrm{out}}}}\cos\frac{\pi (2j+1)i}{2d_{\mathrm{out}}} & \left(1\leq i\leq d_{\mathrm{out}}\right) \end{array}\right.. \end{array}$$

One of the contributions of this study is the finding that combining positional encoding and the DCT matrix with the ArcFace model significantly boosts performance, whereas the standard ArcFace model without these additions performs similarly to the softmax function.

### 4.5. Jeffrey Divergence Regularization

We empirically observed that optimizing the loss function described above is extremely challenging. In particular, the L1 distance cannot be differentiated at 0. Figure 3b illustrates the learning curve for standard optimization using the softmax function model.

To stabilize optimization, we propose including the Jeffrey divergence (JD) as a regularization term. JD is an upper bound of the square of the 1-Wasserstein distance.

**Proposition** **2.** 
*For $B^{⊤}w=1$ and probability vectors $a_{i}$ and $a_{j}$, we have*

$$\begin{array}{c} W_{T}^{2}\left(\mu_{i},\mu_{j}\right)\leq \mathrm{JD}\left(\mathrm{diag}\left(w\right)Ba_{i}∥\mathrm{diag}\left(w\right)Ba_{j}\right), \end{array}$$

*where*

JD(diag(w)Bai∥diag(w)Baj)=KL(diag(w)Bai∥diag(w)Baj)=+KL(diag(w)Baj∥diag(w)Bai)

*is a Jeffrey divergence.*


**Proof.** The following holds if $B^{⊤}w=1$ with the probability vector a (such that $a^{⊤}1=1$).
$$\begin{array}{c} 1^{⊤}\mathrm{diag}\left(w\right)Ba=1. \end{array}$$Then, using Pinsker’s Inequality, we derive the following inequalities:
$$\begin{array}{c} W_{T}\left(\mu_{i},\mu_{j}\right)={∥\mathrm{diag}\left(w\right)Ba_{i}-\mathrm{diag}\left(w\right)Ba_{j}∥}_{1}\leq \sqrt{2\mathrm{KL}(\mathrm{diag}\left(w\right)Ba_{i}∥\mathrm{diag}\left(w\right)Ba_{j})}, \end{array}$$
and
$$\begin{array}{c} W_{T}\left(\mu_{i},\mu_{j}\right)={∥\mathrm{diag}\left(w\right)Ba_{j}-\mathrm{diag}\left(w\right)Ba_{i}∥}_{1}\leq \sqrt{2\mathrm{KL}(\mathrm{diag}\left(w\right)Ba_{j}∥\mathrm{diag}\left(w\right)Ba_{i})}, \end{array}$$Thus,
$$\begin{array}{cc} W_{T}^{2}\left(\mu_{i},\mu_{j}\right) & \leq \mathrm{KL}(\mathrm{diag}\left(w\right)Ba_{i}∥\mathrm{diag}\left(w\right)Ba_{j})+\mathrm{KL}(\mathrm{diag}\left(w\right)Ba_{j}∥\mathrm{diag}\left(w\right)Ba_{i}) \end{array}$$□

This result indicates that minimizing the symmetric KL divergence (i.e., Jeffrey divergence) can minimize the tree-Wasserstein distance. Due to the fact that the Jeffrey divergence is smooth, the computation of the gradient of the upper bound is easier. For presentation, we denote $W_{T}\left(\mu^{\left(1\right)},\mu^{\left(2\right)}\right)=W_{T}\left(a^{\left(1\right)},a^{\left(2\right)}\right)$.

Note that Frogner et al. [53] considered a multilabel classification problem utilizing the regularized Wasserstein loss. They proposed utilizing Kullback–Leibler divergence-based regularization to stabilize training. We derive the Jeffrey divergence from the TWD, and JD regularization includes a simple KL divergence-based regularization as a special case. Moreover, we propose employing JD regularization for SSL frameworks, which have not been extensively studied.

## 5. Experiments

This section evaluates SSL methods with different probability models.

### 5.1. Performance Comparison for SimCLR

For all experiments, we employed the Resnet18 model with an output dimension of ($d_{\mathrm{out}}=256$) and coded all the methods based on a standard SimCLR implementation (https://github.com/sthalles/SimCLR (accessed on 7 July 2023). We used the Adam optimizer and set the learning rate to 0.0003, the weight decay parameter to 1e-4, and temperature τ to 0.07. For the proposed method, we compared two variants of TWD: total variation and ClusterTree (Figure 1). As part of the model evaluation, we assessed the conventional softmax function, attention model (AF), and simplicial embedding (SEM) [10] and set the temperature parameter τ=0.1 for all experiments. For SEM, we set L=16 and V=16.

We also evaluated JD regularization, where we set the regularization parameter λ=0.1 for all experiments. For reference, we compared cosine similarity as a similarity function of SimCLR. For all approaches, we utilized the KNN classifier of the scikit-learn package (https://scikit-learn.org/stable/modules/generated/sklearn.neighbors.KNeighborsClassifier.html (accessed on 7 July 2023)), where the number of nearest neighbor was set to K=50. We utilized the L1 distance for Wasserstein distances and cosine similarity for non-probability-based models. All the experiments were computed on A6000 GPUs. We ran all experiments three times and report the average scores.

Figure 3 illustrates the training loss and top-1 accuracy for the three methods: cosine + real-valued embedding, TV + softmax, and TV + AF (DCT). This experiment revealed that the convergence speed of the loss function was nearly identical across all methods. Regarding the training top-1 accuracy, cosine + real-valued embedding achieves the highest accuracy, followed by the softmax function, and AF (DCT) lags. This behavior is expected because real-valued embeddings offer the most flexibility, followed by softmax, with AF models exhibiting the least freedom. For all methods based on the TWD, JD regularization significantly aids the training process, particularly in the case of the softmax function. However, for AF (DCT), the improvement was relatively small. This is likely because AF (DCT) can also be considered a form of regularization.

Table 1 presents the experimental results for the test classification accuracy using KNN. The first observation is that the simple implementation of the conventional softmax function performs poorly (the performance is approximately 10 points lower) compared to cosine similarity. As expected, AF has only one more layer than the simple softmax model, and performs similarly to softmax. Compared to softmax and AF, AF (PE), and AF (DCT) significantly improve the classification accuracy for the total variation and ClusterTree cases. However, for the ClusterTree case, AF (PE) achieves a better classification performance, whereas the AF (DCT) improvement over the softmax model is limited. In the ClusterTree case, SEM significantly improves with the combination of ClusterTree and regularization. One potential reason of the performance improvement on TV + AF (DCT) combination and ClusterTree + SEM is that AF (DCT) utilizes the orthonormal DCT transform of the learned representation, while both SEM and ClusterTree have structures themselves. This means that each element of the final probability vector $a_{\theta }$ can be uncorrelated for AF (DCT). As a result, the tree structure may not provide significant information, and the total variation (i.e., each leaf node connected to the root node) might be the best fit for the probability representation. Additionally, the cluster-like structure may conflict with the DCT-based representation. In contrast, SEM has an inherent structure and is computed without the DCT transformation (it learns a sum-to-one vector on subtrees). Therefore, the cluster tree structure and SEM can be a good match.

Overall, the proposed method performs better than cosine similarity without real-valued vector embedding when the number of classes is relatively small (i.e., STL10, CIFAR10, and SVHN). By contrast, the performance of the proposed method degrades for CIFAR100, and the results for ClusterTree are particularly poor. As the Wasserstein distance can be minimized even if it cannot overfit, it is natural for the Wasserstein distance to make mistakes when the number of classes is large. Note that the performances for CIFAR100 with simplicial representation degrade both cosine and TWD loss functions, and the performance degradation seems to come from the softmax operation. Moreover, the total variation is a robust measure and learning with total variation is generally designed to create models that are resilient to noise. In our setting, which involves self-supervised learning, it is likely that similar class representations could become mixed, leading to performance degradation. Since the proposed method performs well on CIFAR-10, we believe this could be the reason for the performance issues on larger datasets. To address this, it may be beneficial to use other types of regularizers or larger deep learning models.

Next, we evaluated the Jeffrey divergence regularization. Surprisingly, simple regularization dramatically improves the classification performance of all the probability models. These results support the idea that the main problem with Wasserstein distance-based representation learning is its numerical instability.

Among the methods, the proposed AF (DCT) + JD with total variation achieves the highest classification accuracy, comparable to the cosine similarity result, and achieves more than 10% improvement from the naive implementation with the softmax function. Moreover, all probability model performances with the cosine similarity combination tend to result in a lower classification error than those with the combination of the TWD and probability models. Based on our empirical study, we propose utilizing TWD (TV) + AF models or TWD (ClusterTree) + SEM for representation-learning tasks in probability-based representation learning.

### 5.2. Performance Comparison for SimSiam

Next, we evaluated the performance using a non-contrastive setup. For all experiments, we utilized the Resnet18-Cifar-Variant1 model with an output dimension of ($d_{\mathrm{out}}=2048$) and implemented all methods based on a standard SimSiam framework (https://github.com/PatrickHua/SimSiam). The optimization was performed using the SGD optimizer with a base learning rate of 0.03, weight decay parameter of 0.00005, momentum parameter of 0.9, batch size of 512, and a fixed number of epochs set to 800. For the proposed method, we employed the total variation as a loss function, along with the softmax function and ArcFace model (AF). The temperature parameter τ was set to 0.1 for all experiments. Additionally, we assessed JD regularization with the regularization parameter λ set to 0.1 across all experiments. A100 GPUs were used for all experiments, and each experiment was run three times, with the reported results being the average scores.

We compared the proposed methods, TWDSimSiam (softmax + JD) and TWDSimSiam (AF + JD), with the original SimSiam method which employs cosine similarity loss. Upon examination, we observe that learning the total variation with softmax encounters numerical issues, even with JD regularization (See Figure 4a,c). Conversely, the AF + JD combination proved successful in training the models, as shown in Figure 4b,c. One potential reason for the failure of TWD with softmax is that the total variation can easily become zero because the softmax function lacks normalization. For TWDSimSiam (AF + JD), normalization within the AF model prevents convergence to a poor local minimum. From a performance standpoint as shown in Table 2, the utilization of cosine similarity and total variation (TV) yield comparable results. However, a key contribution of this study is the introduction of a practical approach to enhance the model training stability by incorporating Wasserstein distance, specifically through total variation. This discovery has a potential utility in various SSL tasks.

### 5.3. Effect of Number of Nearest Neighbors

In this section, we assess the performance of the KNN model by varying the number of nearest neighbors and setting *K* to 10 or 50. The results for K=10 are presented in Table 3, and Table 4 illustrates a comparison of the best models across different nearest neighbor values. Our experiments revealed that utilizing K=50 tends to enhance the performance, and the relative order of the results remains consistent, regardless of the number of nearest neighbors.

### 5.4. Effect of the Regularization Parameter for Jeffrey Divergence

In this experiment, we evaluated model performance by varying the regularization parameter, denoted as λ. The results indicate a noteworthy improvement in performance with the introduction of regularization parameters. However, as shown in Table 5, it was observed that the performance did not exhibit significant changes across different values of λ, and setting λ=0.1 yielded favorable results.

## 6. Conclusions

This study investigates SSL using TWD. We empirically evaluate several benchmark datasets and find that a simple combination of the softmax function and TWD performs poorly. To address this, we propose simplicial embedding [10] and ArcFace models [21] as probability models. Moreover, to mitigate the intricacies of optimizing TWD, we incorporate an upper bound on the squared 1-Wasserstein distance as a regularization technique. Overall, the combination of ArcFace and DCT outperforms their cosine similarity counterparts. Finally, we find that the combination of TWD (ClusterTree) and SEM yields favorable performance.

There are several potential future directions for our work. Firstly, improving representation learning for larger classes could involve employing larger models and/or introducing new regularization techniques. Secondly, integrating the proposed probability representation into other SSL models such as DINO [8] could enhance our understanding of model performance across different learning tasks. Lastly, while we have empirically studied self-supervised learning with Wasserstein distance, the theoretical properties remain unclear. Therefore, investigating these theoretical properties represents another promising research direction.

## Figures and Tables

**Figure 1 entropy-26-00939-f001:**
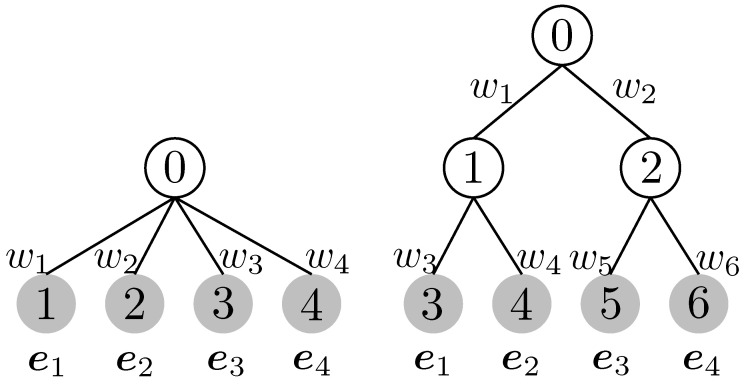
Left tree corresponds to the total variation if we set the weight as $w_{i}=\frac{1}{2},\forall i$. Right tree is a ClusterTree (2 class).

**Figure 2 entropy-26-00939-f002:**
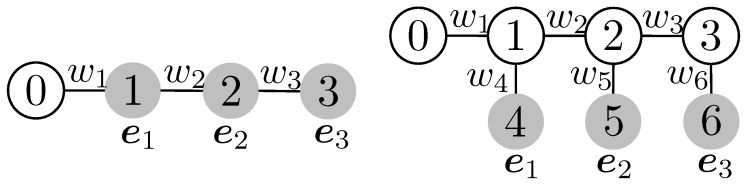
Tree for sliced Wasserstein distance for $N_{\mathrm{leaf}}=3$. The left figure is a chain and the right figure is the tree representation with internal nodes for the chain ($w_{4}=w_{5}=w_{6}=0$).

**Figure 3 entropy-26-00939-f003:**
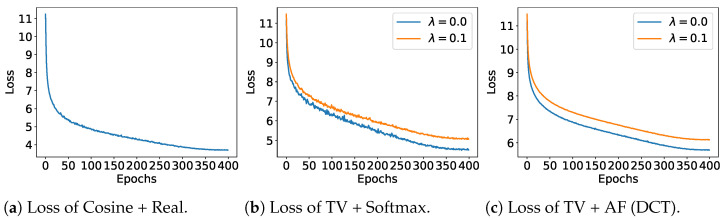
InfoNCE loss and Top-1 (Train) comparisons on STL10 dataset.

**Figure 4 entropy-26-00939-f004:**
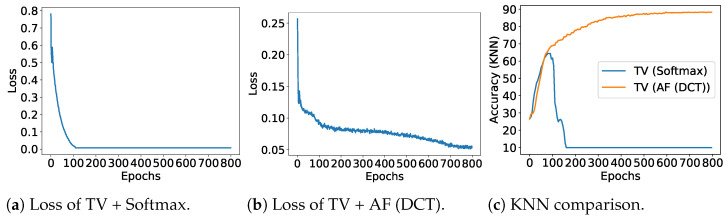
TWD loss for SimSiam models.

**Table 1 entropy-26-00939-t001:** KNN classification result with Resnet18 backbone. In this experiment, we set the number of neighbors as K=50 and computed the averaged classification accuracy over three runs. Note that the Wasserstein distance with ($B=I_{d_{\mathrm{out}}}$) is equivalent to total variation.

Similarity	Prob Model	STL10	CIFAR10	CIFAR100	SVHN
Cosine	N/A	**75.77** **± 0.47**	**67.39** **± 0.46**	**32.06** **± 0.06**	76.35 ± 0.39
	Softmax	70.12 ± 0.04	63.20 ± 0.23	26.88 ± 0.26	74.46 ± 0.62
	SEM	71.33 ± 0.45	61.13 ± 0.56	26.08 ± 0.07	74.28 ± 1.13
	AF (DCT)	72.95 ± 0.31	65.92 ± 0.65	25.96 ± 0.13	**76.51** **± 0.24**
TWD (TV)	Softmax	65.54 ± 0.47	59.72 ± 0.39	26.07 ± 0.19	72.67 ± 0.33
	SEM	65.35 ± 0.31	56.56 ± 0.46	24.31 ± 0.43	73.36 ± 1.19
	AF	65.61 ± 0.56	60.92 ± 0.42	26.33 ± 0.42	75.01 ± 0.32
	AF (PE)	71.71 ± 0.17	64.68 ± 0.33	26.38 ± 0.37	76.44 ± 0.45
	AF (DCT)	73.28 ± 0.27	67.03 ± 0.24	25.85 ± 0.39	77.62 ± 0.40
	Softmax + JD	72.64 ± 0.27	67.08 ± 0.14	**27.82** **± 0.22**	77.69 ± 0.46
	SEM + JD	71.79 ± 0.92	63.60 ± 0.50	26.14 ± 0.40	75.64 ± 0.44
	AF + JD	72.64 ± 0.37	67.15 ± 0.27	27.45 ± 0.37	78.00 ± 0.15
	AF (PE) + JD	74.47 ± 0.10	67.28 ± 0.65	27.01 ± 0.39	78.12 ± 0.48
	AF (DCT) + JD	**76.28** **± 0.07**	**68.60** **± 0.36**	26.49 ± 0.24	**79.70** **± 0.23**
TWD (Clus)	Softmax	69.15 ± 0.45	62.33 ± 0.40	24.47 ± 0.40	74.87 ± 0.13
	SEM	72.88 ± 0.12	63.82 ± 0.32	22.55 ± 0.28	77.47 ± 0.92
	AF	70.40 ± 0.40	63.28 ± 0.57	24.28 ± 0.15	75.24 ± 0.52
	AF (PE)	72.37 ± 0.28	65.08 ± 0.74	23.33 ± 0.35	76.67 ± 0.26
	AF (DCT)	71.95 ± 0.46	65.89 ± 0.11	21.87 ± 0.19	77.92 ± 0.24
	Softmax + JD	73.52 ± 0.16	66.76 ± 0.29	**24.96** **± 0.07**	77.65 ± 0.53
	SEM + JD	**75.93** **± 0.14**	**67.68** **± 0.46**	22.96 ± 0.28	**79.19** **± 0.53**
	AF + JD	73.66 ± 0.23	66.61 ± 0.32	24.55 ± 0.14	77.64 ± 0.19
	AF (PE) + JD	73.92 ± 0.57	67.00 ± 0.13	23.83 ± 0.42	77.87 ± 0.29
	AF (DCT) + JD	74.29 ± 0.30	67.50 ± 0.49	22.89 ± 0.12	78.31 ± 0.72

**Table 2 entropy-26-00939-t002:** SimSiam evaluation with CIFAR10 dataset.

Similarity	Probability Model	Linear Classifier
Cosine	N/A	91.13 ± 0.14
TWD (TV)	Softmax + JD	9.99 ± 0.00
	AF (DCT) + JD	90.60 ± 0.02

**Table 3 entropy-26-00939-t003:** KNN classification result with Resnet18 backbone. In this experiment, we set the number of neighbors as K=10 and computed the averaged classification accuracy over three runs. Note that the Wasserstein distance with ($B=I_{d_{\mathrm{out}}}$) is equivalent to a total variation.

Similarity	Prob Model	STL10	CIFAR10	CIFAR100	SVHN
Cosine	N/A	**75.44** **± 0.21**	**66.96** **± 0.45**	**31.63** **± 0.25**	74.71 ± 0.31
	Softmax	71.25 ± 0.30	63.80 ± 0.48	26.18 ± 0.36	73.06 ± 0.47
	SEM	71.34 ± 0.31	61.26 ± 0.42	25.40 ± 0.06	73.41 ± 0.95
	AF (DCT)	72.15 ± 0.53	65.52 ± 0.45	24.93 ± 0.24	**75.68** **± 0.13**
TWD (TV)	Softmax	63.42 ± 0.24	59.03 ± 0.58	24.95 ± 0.31	70.87 ± 0.29
	SEM	63.72 ± 0.17	55.57 ± 0.35	23.40 ± 0.36	71.69 ± 0.75
	AF	63.97 ± 0.05	59.96 ± 0.44	25.29 ± 0.17	73.44 ± 0.35
	AF (PE)	71.04 ± 0.37	64.28 ± 0.14	25.71 ± 0.20	75.70 ± 0.42
	AF (DCT)	72.75 ± 0.11	67.01 ± 0.03	24.95 ± 0.17	76.98 ± 0.44
	Softmax + JD	72.05 ± 0.30	66.61 ± 0.20	26.91 ± 0.19	76.65 ± 0.56
	SEM + JD	70.73 ± 0.89	62.75 ± 0.61	24.83 ± 0.27	74.71 ± 0.43
	AF + JD	71.74 ± 0.19	66.74 ± 0.20	**26.68** **± 0.35**	77.10 ± 0.04
	AF (PE) + JD	74.10 ± 0.20	66.82 ± 0.36	26.17 ± 0.00	77.55 ± 0.50
	AF (DCT) + JD	**76.24** **± 0.22**	**68.62** **± 0.40**	25.70 ± 0.14	**79.28** **± 0.22**
TWD (Clust)	Softmax	67.95 ± 0.42	61.59 ± 0.29	23.34 ± 0.26	73.88 ± 0.05
	SEM	72.43 ± 0.11	63.63 ± 0.42	21.29 ± 0.28	77.04 ± 0.77
	AF	69.09 ± 0.05	62.49 ± 0.45	22.56 ± 0.25	74.31 ± 0.40
	AF (PE)	72.08 ± 0.07	64.56 ± 0.31	22.51 ± 0.29	75.98 ± 0.23
	AF (DCT)	71.64 ± 0.15	65.51 ± 0.36	21.04 ± 0.10	77.59 ± 0.25
	Softmax + JD	73.07 ± 0.13	66.38 ± 0.27	**23.97** **± 0.11**	76.82 ± 0.50
	SEM + JD	**75.50** **± 0.15**	**67.44** **± 0.10**	21.90 ± 0.19	**78.91** **± 0.30**
	AF + JD	72.70 ± 0.08	66.12 ± 0.26	23.50 ± 0.21	76.92 ± 0.06
	AF (PE) + JD	73.66 ± 0.47	66.58 ± 0.01	22.86 ± 0.02	77.44 ± 0.30
	AF (DCT) + JD	73.79 ± 0.12	67.34 ± 0.38	21.96 ± 0.34	78.00 ± 0.60

**Table 4 entropy-26-00939-t004:** KNN classification accuracy with different number of neighbors.

Similarity	*K*	STL10	CIFAR10	CIFAR100	SVHN
TWD (TV)	10	76.24 ± 0.22	68.62 ± 0.40	25.70 ± 0.14	79.28 ± 0.22
	50	76.28 ± 0.07	68.60 ± 0.36	26.49 ± 0.24	79.70 ± 0.23

**Table 5 entropy-26-00939-t005:** KNN classification result with Resnet18 backbone. In this experiment, we set the number of neighbors as K=50 and computed the averaged classification accuracy over three runs.

Similarity Function	λ	STL10	CIFAR10	CIFAR100	SVHN
TWD (TV)	0.0	73.28 ± 0.27	67.03 ± 0.24	25.85 ± 0.39	77.62 ± 0.40
	0.1	76.28 ± 0.07	**68.60** **± 0.36**	**26.49** **± 0.24**	79.70 ± 0.23
	0.2	77.40 ± 0.17	68.48 ± 0.11	25.59 ± 0.16	79.67 ± 0.26
	0.3	**77.67** **± 0.06**	68.26 ± 0.51	24.21 ± 0.35	**79.91** **± 0.42**

## Data Availability

All the data used in the study is publicly accessible.

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
