# Peer review of "An Empirical Study of Self-Supervised Learning with Wasserstein Distance"

_entropy, 2024, doi:10.3390/e26110939_

Round 1
Reviewer 1 Report
Comments and Suggestions for Authors
The authors employ the 1-Wasserstein distance on a tree structure (TWD) to define similarity in the self-supervised learning (SSL) Thus, this study empirically investigates a strategy for optimizing the SSL with the Wasserstein distance and finds a stable training procedure. They evaluate the combinations of two types of TWD (total variation and ClusterTree) and several probability models and propose Jeffrey divergence-based regularization to stabilize optimization. The empirical experiment in the paper shows some interesting results and concludes that some combinations outperform cosine- similarity based representation learning. A good measure of similarity/dissimilarity in a learning model is a key ingredient. Wasserstein distance often provides a better understanding of the similarity of digital images and similar objects, particularly in classifications. The research in the paper offers a valuable try in this aspect.
The experiments show that, in average, TWD(TV) on AF(DCT)+JD gives more satisfactory results while TWD(Clus) has better performance on SEM_JD model. Are there intrinsic reasons? On CIFAR100, Cosine similarity seems still better than TWD. There are lots further researches on the model analysis and selections.
Author Response
Thank you for your valuable comments.
> The experiments show that, in average, TWD(TV) on AF(DCT)+JD gives more satisfactory results while TWD(Clus) has better performance on SEM_JD model. Are there intrinsic reasons?
Thank you for the comment. We also found that the behavior is really interesting. However, we are still not sure why TWD (Clust) gives better performance on SEM JD.
One potential reason of the performance improvement on TV + AF (DCT) combination and ClusterTree + SEM is that AF (DCT) utilizes the orthonormal DCT transform of the learned representation, while both SEM and ClusterTree have structures themselves. This means that each element of the final probability vector a_\theta can be uncorrelated for AF (DCT). As a result, the tree structure may not provide significant information, and the total variation (i.e., each leaf node connected to the root node) might be the best fit for the probability representation. Additionally, the cluster-like structure may conflict with the DCT-based representation. In contrast, SEM has an inherent structure and is computed without the DCT transformation (it learns a sum-to-one vector on subtrees). Therefore, the cluster tree structure and SEM can be a good match. We have added this discussion about the differences in the revised manuscript
> On CIFAR100, Cosine similarity seems still better than TWD. There are lots further researches on the model analysis and selections.
Thank you for pointing this out. Indeed, we observed that cosine similarity performs better than the proposed method on larger class classifications. In general, the purpose of this paper is to empirically investigate the behavior of representation learning with the Wasserstein distance, and we aim to identify potential issues with learning the Wasserstein distance. CIFAR-100 is one of the challenges we encountered during the study. We included this as future work in the conclusion.
The performances for CIFAR100 with simplicial representation degrade both Cosine and TWD loss functions, and the performance degradation seems to come from the softmax operation. Moreover, the total variation is a robust measure and learning with total variation is generally designed to create models that are resilient to noise. In our setting, which involves self-supervised learning, it is likely that similar class representations could become mixed, leading to performance degradation. Since the proposed method performs well on CIFAR-10, we believe this could be the reason for the performance issues on larger datasets. To address this, it may be beneficial to use other types of regularizers or larger deep learning models. We also added this information and discussion to the experiments and the future work sections.
Reviewer 2 Report
Comments and Suggestions for Authors
Following are the comments and suggestions for improving the manuscript.
- "In contastive learning": Typo here, should be "contrastive learning.”
- SimCLR and InfoNCE Loss: It might be worth adding a brief explanation of the InfoNCE loss, especially for readers who might not be familiar with it.
- Throughout the manuscript, terms like ArcFace, DCT, simplicial embedding, and TWD are introduced without sufficient explanation. A reader unfamiliar with these concepts may struggle to follow. Ensure that key methods and models are clearly explained, particularly in the methods and results sections. Providing a background or brief definitions, perhaps in the introduction or method sections, will make the paper more accessible.
- The manuscript presents multiple advanced methods, but the novelty of the proposed approach isn’t highlighted. Is it the combination of methods that provides new insights or a new formulation of one of these methods? Make the key novel contribution of your work stand out more clearly.
Proofreading the manuscript can help eliminate minor typos and grammar issues.
Author Response
Thank you for your valuable comments.
> "In contastive learning": Typo here, should be "contrastive learning.”
Thank you for pointing out the typo. We fixed it. We also proofread the paper and fixed other typos.
>SimCLR and InfoNCE Loss: It might be worth adding a brief explanation of the InfoNCE loss, especially for readers who might not be familiar with it.
We added more detailed information about the InfoNCE loss.
> Throughout the manuscript, terms like ArcFace, DCT, simplicial embedding, and TWD are introduced without sufficient explanation. A reader unfamiliar with these concepts may struggle to follow. Ensure that key methods and models are clearly explained, particularly in the methods and results sections. Providing a background or brief definitions, perhaps in the introduction or method sections, will make the paper more accessible.
Thank you for the suggestions. In the revised version, we added more explanation with some examples. We hope this makes the paper more accessible to broader audience.
> The manuscript presents multiple advanced methods, but the novelty of the proposed approach isn’t highlighted. Is it the combination of methods that provides new insights or a new formulation of one of these methods? Make the key novel contribution of your work stand out more clearly.
We have expanded the contribution section in the introduction. The main contribution of this paper is the first-time combination of the Wasserstein distance with self-supervised learning. We then identify the challenges associated with using the Wasserstein distance in SSL and provide simple solutions to stabilize the learning process.
Round 2
Reviewer 2 Report
Comments and Suggestions for Authors
The authors addressed the comments.